# SpatialRGPT: Grounded Spatial Reasoning in Vision-Language Models

**An-Chieh Cheng[1], Hongxu Yin[2], Yang Fu[1], Qiushan Guo[2], Ruihan Yang[1],**
**Jan Kautz[2], Xiaolong Wang[1,2], Sifei Liu[2]**
[1]UC San Diego, [2]NVIDIA

## Ⓐ Relative Relation

Can you confirm if ④ has a greater width compared to ⑥?

In fact, ④ might be narrower than ⑥.

Which of these four, ③,④,⑤,⑥ is the tallest?

⑥ is the tallest.

From the viewer's perspective, is the position of ⑥ closer than that of ③?

No, ③ is closer.

## Ⓑ Metric Measurement

If you are at ③, where will you find ⑦?

⑦ is at 1 o'clock from ③.

What is the height of ④?

④ is 1.38 meters tall.

How far apart are ② and ③ horizontally?

They are 15.33 meters apart horizontally.

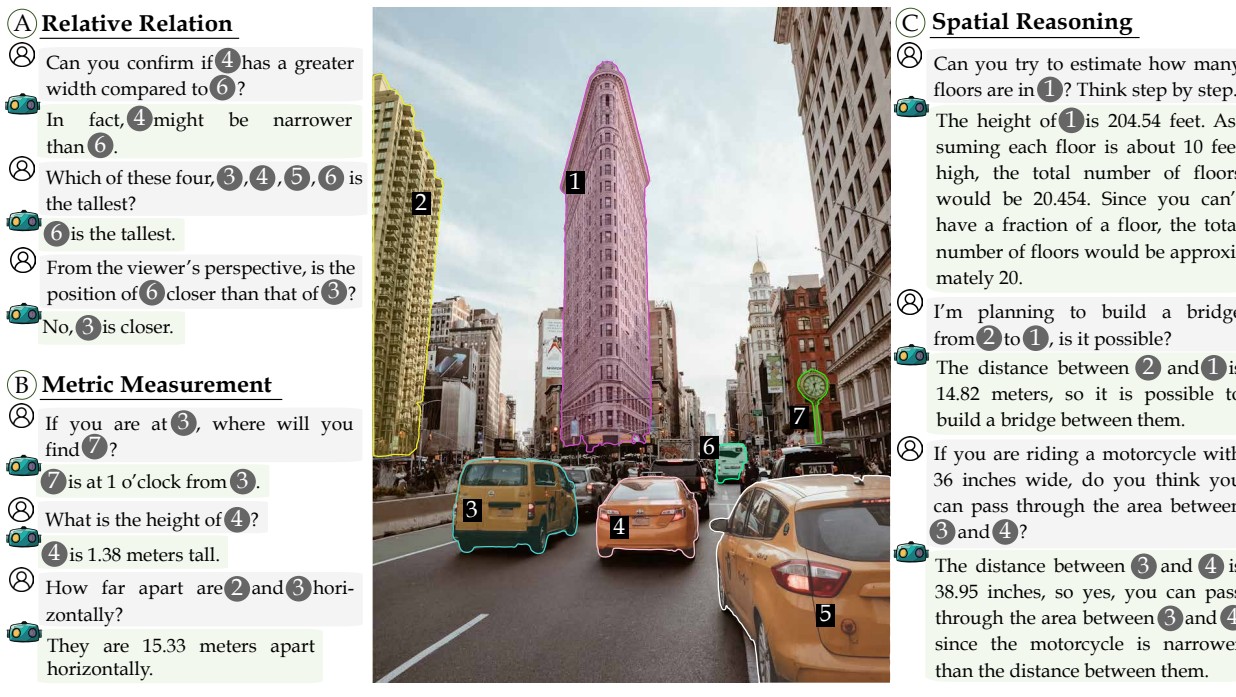

## Ⓒ Spatial Reasoning

Can you try to estimate how many floors are in ①? Think step by step.

The height of ① is 204.54 feet. Assuming each floor is about 10 feet high, the total number of floors would be 20.454. Since you can't have a fraction of a floor, the total number of floors would be approximately 20.

I'm planning to build a bridge from ② to ①, is it possible?

The distance between ② and ① is 14.82 meters, so it is possible to build a bridge between them.

If you are riding a motorcycle with 36 inches wide, do you think you can pass through the area between ③ and ④?

The distance between ③ and ④ is 38.95 inches, so yes, you can pass through the area between ③ and ④ since the motorcycle is narrower than the distance between them.

## Abstract

Vision Language Models (VLMs) have demonstrated remarkable performance in 2D vision and language tasks. However, their ability to reason about spatial arrangements remains limited. In this work, we introduce Spatial Region GPT (SpatialRGPT) to enhance VLMs' spatial perception and reasoning capabilities. SpatialRGPT advances VLMs' spatial understanding through two key innovations: (i) a data curation pipeline that enables effective learning of regional representation from 3D scene graphs, and (ii) a flexible "plugin" module for integrating depth information into the visual encoder of existing VLMs. During inference, when provided with user-specified region proposals, SpatialRGPT can accurately perceive their relative directions and distances. Additionally, we propose SpatialRGBT-Bench, a benchmark with ground-truth 3D annotations encompassing indoor, outdoor, and simulated environments, for evaluating 3D spatial cognition in VLMs. Our results demonstrate that SpatialRGPT significantly enhances performance in spatial reasoning tasks, both with and without local region prompts. The model also exhibits strong generalization capabilities, effectively reasoning about complex spatial relations and functioning as a region-aware dense reward annotator for robotic tasks. Code, dataset, and benchmark are released at https://www.anjiecheng.me/SpatialRGPT.

38th Conference on Neural Information Processing Systems (NeurIPS 2024).

# 1  Introduction

Understanding spatial arrangements in both 2D [1, 2] and 3D [3] spaces is crucial for accurately interpreting complex visual environments. Despite the impressive advancements in Vision Language Models (VLMs) across a variety of tasks such as image classification [4], captioning [5], object detection [6], video understanding [7], and document parsing [8], etc., these models still face significant challenges with spatial reasoning. This includes difficulties [9, 10, 11] in distinguishing simple spatial concepts like "left" and "right," "above" and "below," as well as more complex relationships such as "behind" and "in front," "inside" and "outside," and "near" and "far." The ability to comprehend and reason about these spatial relationships is fundamental not only for visual understanding, but also for enabling practical applications in fields like robotics [12, 13] and augmented reality [14], where precise spatial awareness is crucial for tasks such as navigation [15], manipulation [12], and interaction with real-world environments [16].

Recently, several works [11, 17, 18] has advanced VLMs' spatial reasoning capabilities by introducing a comprehensive data generation pipeline that enables large-scale training with spatially-aware visual question answering (VQA) tasks. This approach is based on the hypothesis that the limited spatial reasoning capabilities of current VLMs are due to a lack of 3D/2D spatial knowledge in their training data. However, two critical challenges remain. First, effective spatial reasoning requires VLMs to accurately parse regional information, particularly the regions of object instances, whereas most existing VLMs are primarily designed to understand the global context of an image. When an image contains numerous instances, it becomes challenging to prompt the model to reason about the spatial relations between specific instances. This is because most VLMs function as global image parsers and do not support specifying regions for which users want to understand spatial relationships. Second, accurately perceiving spatial relations such as direction and distance cannot rely solely on RGB pixel data. Thus, the architecture needs to incorporate 3D inputs, such as depth information.

In this work, we propose SpatialRGPT, leveraging a data curation pipeline, along with a region and 3D-aware visual encoder architecture to improve the spatial reasoning capability of VLMs.

Our data pipeline automatically generates 3D, region-aware annotations from 2D images at scale by constructing a 3D scene graph for each image, where nodes represent object instances and edges denote spatial relationships. This is achieved through three scalable components: (i) open-vocabulary detection and segmentation for instance extraction, (ii) metric depth estimation, and (iii) camera calibration for projecting objects into 3D space. These scene graphs are subsequently transformed into region-aware spatial QA tasks using both template-based and large language model (LLM)-based approaches. This dual approach provides region-based VLMs with the necessary spatial knowledge and advanced reasoning capabilities to interpret complex environments. We use the collected data to train SpatialRGPT. While SpatialRGPT is designed to support region prompts, it effectively avoids the ambiguity issues found in SpatialVLM. In SpatialVLM, multiple similar objects in an image can confuse caption labels. In contrast, our pipeline naturally handles these scenarios without requiring carefully crafted rules or extensive post-processing.

Similar to RGPT [19], SpatialRGPT introduces a region representation module that allows region proposals to be included as additional inputs alongside the image. This approach enables the LLM to leverage both regional and global contexts, allowing the model to reason about relationships between local regions while maintaining an understanding of the overall scene. In addition, we propose a novel architecture that features a flexible "plugin" module for integrating relative-depth information into the visual encoder of existing VLMs. This design allows a pre-trained visual encoder to optionally learn additional depth representation while still functioning effectively when depth inputs are absent. Our experiments demonstrate that this design can substantially improve the spatial reasoning capabilities compared to VLMs that only use RGB images as input. Furthermore, we highlight practical applications enabled by SpatialRGPT, such as serving as a region-aware dense reward annotator and a stand-alone complex spatial reasoner. Our work has four main contributions:

1. We present SpatialRGPT, a framework that enhances region-level spatial reasoning in VLMs by enabling effective representation of regional information and acquisition of spatial knowledge. Our novel architecture also integrates depth information flexibly, significantly improving 3D perception and analysis.
2. To facilitate model training, we introduce a scalable data pipeline that constructs region-aware spatial reasoning QAs from existing datasets. With the pipeline, we create the Open Spatial Dataset (OSD), encompassing 8.7M spatial concepts grounded in 5M unique regions.

3. To address the absence of a benchmark for evaluating spatial cognition in VLMs, we present SpatialRGPT-Bench, a comprehensive benchmark based on ground-truth 3D annotations that span indoor, outdoor, and simulated environments.

4. We demonstrate downstream applications of SpatialRGPT. Leveraging SpatialRGPT's region capabilities, we develop a region-aware dense reward annotator for robotics. Additionally, we show that SpatialRGPT can function as a stand-alone complex spatial reasoner, as well as its capacity to perform multi-hop reasoning.

## 2   Related work

**Spatial Reasoning via Large Language Models.**   Recently, there has been a significant push to obtain spatial reasoning capabilities using LLMs. Initiatives [20, 21] have focused on reconstructing scenes from multi-view images, such as point clouds or neural fields, and enhancing these representations with dense semantic features. The resulting 3D representation and dense features are then integrated into an LLM. However, multi-view images are not always available, and constructing a scene explicitly with dense semantic features is resource-intensive. Additionally, the modal gap between 3D representations and language often results in decreased performance. ConceptGraph [22] avoids directly incorporating 3D representations into LLMs. Instead, it constructs a scene graph and integrates this with the LLM. Yet, recent studies [10] indicate that LLMs struggle to utilize coordinate information effectively when presented in text, which can undermine their ability to understand and reason about spatial relationships. Our research is most aligned with SpatialVLM [17], which uses 2D VLMs to understand spatial relationships and metric distances. Unlike the above approaches, the spatial understanding is encoded implicitly. The VLM directly handles the spatial relationship problem without an explicit 3D representation or scene graph. However, SpatialVLM relies on language descriptions of objects as input, while LLMs can already resolve some spatial queries even without visual data [23]. The responses can be inferred directly from the questions or derived from the world knowledge embedded in LLMs. This reliance on textual cues suggests that the training may not effectively teach VLMs to learn spatial reasoning from visual data. Additionally, SpatialVLM lacks the capability to specify regions precisely. This is particularly problematic in real-world scenarios where describing ambiguous locations or objects in language can be challenging.

**Region-level Visual Language Models.**   KOSMOS-2 [24], Shikra [25], MiniGPT-2 [26], CogVLM [27], SPHINX [28], and LLaVA [29] have enabled MLLMs to achieve region-based image understanding. However, these methods provide region information in textual form, such as bounding box coordinates. This method heavily depends on the language decoder to understand the position. In contrast, VisionLLM [30], GPT4RoI [31], [32], and Ferret [33, 34], along with GLaMM [35], use spatial boxes with ROI-aligned features to map region-level features into the LLM word embedding space. However, bounding boxes can include unwanted background features, leading to inaccurate alignment between region descriptions and text, which complicates spatial reasoning. Recently, RegionGPT [19] and Osprey [36] have introduced visual spatial-aware modules that can directly extract pixel-level features. These models support using input masks that can accommodate regions of any shape. Despite these advancements, none of these approaches specifically focus on enhancing spatial reasoning at the region level in VLMs. Our framework is based on RegionGPT's ability to process pixel-level inputs, with the aim of deepening spatial reasoning within region VLMs.

## 3   Method

SpatialRGPT is a powerful multimodal language model adept at understanding both 2D and 3D spatial arrangements. It can process any region proposal, such as boxes or masks, and provide answers to spatial reasoning questions. While effective training dataset is the key to learn spatial-aware region representation, we introduce: (i) how to build 3D scene Graph from a single image, in Sec. 3.1, and (ii) how to facilitate visual representation learning from these scene graphs in Sec. 3.2. We propose a novel SpatialRGPT visual encoder architecture that flexibly leveraging monocular depth information into an existing 2D VLM, in Sec. 3.3, with training detail explained in Sec. 3.1.

### 3.1   3D Scene Graph from Single 2D Images

Our scene graph construction pipeline (Figure 1) begins with a filtering process to remove any unsuitable images (Appx. F.1). Using open-vocabulary models, we identify and ground candidate objects, followed by lifting them into 3D space using metric depth estimation and camera calibration. We then process the point clouds (Appx. F.3) to construct the final 3D scene graph.

**Open-Vocabulary Detection & Segmentation.**   Segmenting objects is the initial stage of building a scene graph. Our models must satisfy two criteria: (i) object descriptions, e.g., class labels, should

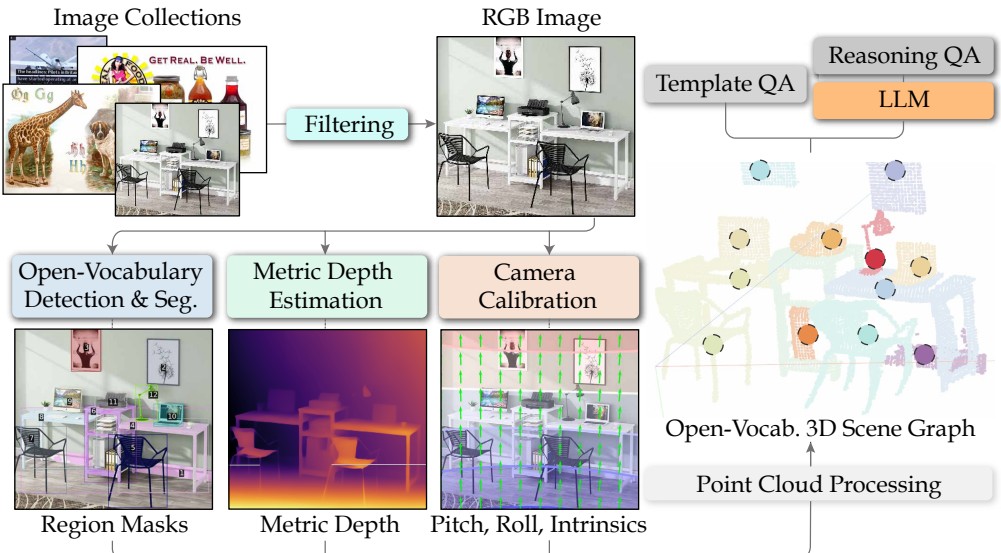

Figure 1: 3D scene graph construction via automatic data curation pipeline.

adhere to an open-world setting for better generalization; (ii) mask proposals need to be highly accurate, ensuring precise contour outlines. This precision is crucial, as even small deviations can lead to significant inaccuracies in the resulting 3D bounding boxes. To this end, we first employ an open-vocabulary image tagging model [37] to identify all the object classes present in the image. Next, we use GroundingDino [38], an open-vocabulary 2D detector to determine the corresponding object bounding boxes. Finally, we apply segmentation models [39] to refine these bounding boxes into precise masks. We do not use existing dataset annotations since they either fall short due to vocabulary limitations, or use polygon annotations [40] or compressed masks [41] for segmentation.

**Metric Depth Estimation.** Several studies have explored the recovery of metric depth from a single image. The main challenge is to address the scale ambiguity, and one common approach [42, 43] is to use relative depth along with metric heads fine-tuned on specific metric datasets. However, these methods may tend to overfit the depth scale for particular datasets such as KITTI [44] or NYU [45], which makes them less robust for in-the-wild images. Recently, Metric3Dv2 [46] takes focal length as input and is trained end-to-end to predict metric depth and surface normals. The model is trained jointly on diverse indoor and outdoor scenes, making it less prone to overfitting to the depth distribution of specific datasets. We adopt Metric3Dv2 as our metric depth estimator and found that Metric3Dv2 together with WildCamera [47]'s camera intrinsic, is robust for images taken in real-world settings. Additionally, thanks to the joint depth-normal optimization training in Metric3Dv2, the recovered geometry is improved particularly around object edges.

**Camera Calibration.** Camera calibration includes (i) intrinsic estimation to back-project depth maps to 3D point clouds, and (ii) scene canonicalization to ensure that scene relations are described in a shared space. To estimate the camera intrinsic, we use the WildCamera model [47], which estimates four DoF intrinsic parameters (focal point and focal length in two dimensions). This model excels in real-world scenarios due to its scale-awareness and ability to detect image cropping. To convert the camera coordinates of the point cloud into a canonicalized geodetic coordinate system for each scene, we leverage PerspectiveFields [48], which provides per-pixel up-vectors and latitude values that can be transformed into camera extrinsics, such as pitch and roll. Using these, we derive a rotation matrix to convert the point cloud from camera coordinates to geodetic coordinates. We note that while SpatialVLM [17] uses surface segmentation (e.g., "floor," "tabletop") to identify a horizontal plane and then uses the normal axis of this plane to align the point cloud to the horizontal plane, this approach is limited by the presence of specific classes, such as floors or tables. Additionally, the plane segmentation may fail if there are not enough points for RANSAC.

**Constructing 3D Scene Graph.** The 3D scene graph is a collection of tuples where the nodes represent specific 3D object instances, and the edges represent the spatial relationships between the nodes. Each node is defined by the object's `class`, `width`, and `height` in metric scale. To create the node, we start by using the instance mask to deproject the object points from the depth map. Then, we perform canonicalization and denoising, and build 3D axis-aligned bounding boxes for each object. With the 3D bounding box, we calculate the width and height of the objects in

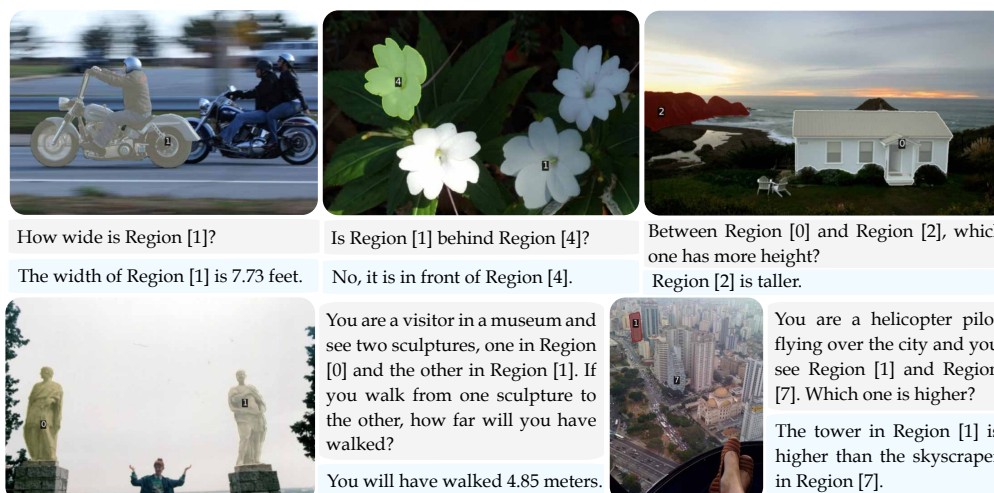

Figure 2: Example data entries from our Open Spatial Dataset. The first row contains template-based QAs, and the second row shows LLM-based entries.

real-world units. The edges represent the spatial relationships between the nodes within two types of relations: relative and metric. Relative relations contain `left`, `right`, `above`, `below`, `behind`, `front`, `wide`, `thin`, `tall`, `short`, `big`, and `small`. Metric relations include `direction`, `direct distance`, `horizontal distance`, and `vertical distance` between the two objects. We then traverse all the object nodes and use the point cloud centroids and bounding boxes to calculate their spatial relationships.

### 3.2 Learning Spatial-aware VLMs from 3D Scene Graph

In this section, we discuss converting the constructed 3D scene graph into textual representations for VLM training. One simple approach is through template-based methods via predefined handcrafted instructions. However, this approach limits the diversity of instructions and hinder the model's reasoning capabilities. Thus, we employ additional complex QAs to enhance the model's reasoning ability. Our results in Figure 4 show that blending these two types of data can lead to a generalized and complex spatial reasoning model.

**Template-based Question Answering.** These QAs serve as the foundation for learning basic spatial knowledge. We extract information about node attributes such as width and height, as well as relative and metric relations from the edge attributes. We create both qualitative and quantitative templates to generate questions and answers for each type of attribute, using entities in the form of `Region [X]`. This approach results in examples shown in the first row of Figure 2. We provide detailed templates for each attribute in Appx. F.4.

**LLM-based Complex Reasoning Question Answering.** We employ Llama3-70B to generate complex spatial reasoning questions to enhance the model's spatial reasoning capabilities. One approach is to input the scene graph directly into the LLMs. However, LLMs struggle to utilize 3D coordinate information effectively [10], so we opt for an alternative approach. We first construct spatial descriptions in a language format. Similar to the template-based approach, we extract attributes from the scene graph and then construct template-based spatial descriptions based on these attributes. We combine the spatial descriptions and the region tags as inputs to the LLM. The LLM is then tasked with creating a complex reasoning question and answer that is based on the description and matches the context. Examples of LLM-generated QAs are shown in the second row of Figure 2. Our LLM prompts for generating QAs are provided in Appx. F.5.

We use our automated annotation pipeline to annotate images from the OpenImages [49] dataset, which covers a wide range of subjects and is of high resolution. The resulting Open Spatial Dataset (OSD) contains 1M unique images and 5M open-vocabulary regions, each associated with a bounding box and segmentation mask. Furthermore, the dataset includes 8M template-based QAs and 700K LLM-based QAs.

### 3.3 VLM Architecture

An overview of SpatialRGPT's VLM architecture is shown in Figure 3. SpatialRGPT consists of a visual encoder (Appx. G.1) to encode vision features, a region-feature extractor [19] to obtain region-level embeddings (Appx. G.2), linear connectors (Appx. G.3) to project multi-modal embeddings into

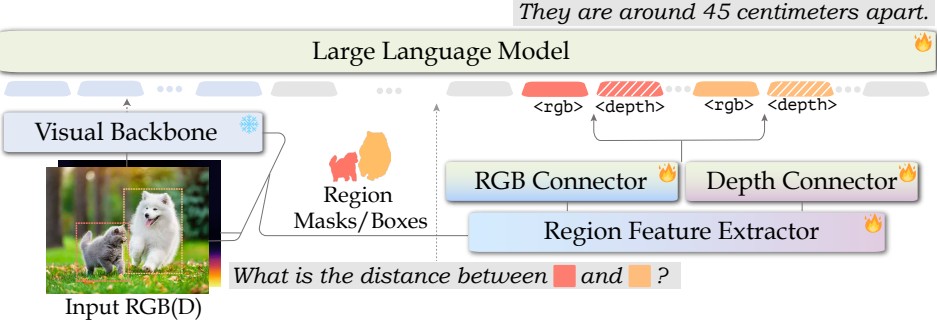

Figure 3: An architecture overview of Spatial RGPT. ❄ 🔥 denotes freezed/trainable parameters.

the word embedding space, and a large language model using LLaMA2-7B for language processing. In this section, we will explain why and how we incorporate depth information into SpatialRGPT, as well as how SpatialRGPT handles tokenizations.

**Plugin Module for Relative-depth Inputs.** VLMs that learn solely from RGB pixels are ineffective for 3D perception tasks. Direct learning from 3D data (e.g., point clouds), presents challenges due to issues with scale and diversity. To bridge this gap, we propose using relative depth, which can be obtained through off-the-shelf models [43], to provide additional 3D information alongside RGB as input to our network. Our goal is to elicit geometric reasoning capability through depth guidance. However, this goal is non-trivial. Most VLM's visual encoders are typically only trained with text and 2D images, and simply concatenating RGB and depth features may negatively impact performance. To address this, we introduce an add-on module that seamlessly incorporates the depth information. We use the same image encoder to process the depth map and generate depth feature maps. Then, we employ an additional depth-to-language connector to project the features into the language domain. The depth connector's weights are trained only on spatial-related QAs. This flexible design allows the 2D visual encoder to leverage additional depth representation while still functioning when depth inputs are not presented, thus avoiding the need for a vast amount of training data.

**Tokenization and Prompt Format.** We generate multi-turn conversation data following [29, 19] for each image and make the image the initial input for the first instruction, providing contextual information. Specifically, we incorporate a prefix prompt: "`<image>\n`". The `<image>` is a special token that acts as a placeholder, which would be replaced by the image-level embedding from the vision encoder. When specific mask regions are mentioned in the user input, we use special tokens `<region>` and `<depth>` as placeholders. Each region token will be substituted with the corresponding region RGB embedding and depth embedding. All image-level, region-level RGB/depth tokens and text tokens are interleaved and fed as the input to the LLM for an auto-regressive generation.

### 3.4 Training and Inference Paradigm

SpatialRGPT training includes three stages [50]: (i) Connector Feature Alignment, (ii) Visual Language Pre-training, and (iii) Visual Instruction-tuning. During the first stage, CC3M image-caption pairs are used to pretrain the RGB connector as [29, 51, 52]. In the second stage, the visual language corpus from MMC4 [53] and COYO [54], along with region understanding datasets from [19] and our OSD dataset, are used to pretrain the LLM and connectors (Figure 3). Finally, at stage three, we fine-tune all weights of the VLM on visual language instruction-following datasets, using a combination of the instruction tuning dataset from [29], region-level instruction tuning data [19], and our OSD dataset. Detailed data blend of the visual instruction data is in Appx. H.1. For training region-level data and our OSD, we randomly sample from different modalities (e.g., box, mask) for each sample to ensure the model is versatile to the input modality. At inference time, SpatialRGPT can take both boxes or masks as input. For the results shown in the main paper, if the segmentation is available, we use the mask; if not, we use the box provided and apply SAM to segment the corresponding mask.

## 4 Experiments

We evaluate the effectiveness of our proposed SpatialRGPT in three aspects: (1) spatial reasoning benchmarks (Section 4.1), (2) standard vision-language benchmarks (Section 4.2), and (3) real-world applications (Section 4.3).

| | Below/Above | Left/Right | Big/Small | Tall/Short | Wide/Thin | Behind/Front | Avg. |
|---|---|---|---|---|---|---|---|
| GPT-4 [55] | 64.16 | 42.85 | 42.85 | 61.60 | 61.60 | 49.09 | 57.83 |
| GPT-4V [55] | 63.34 | 46.67 | 64.15 | 60.71 | 68.26 | 45.45 | 58.14 |
| LLaVA-v1.6-34B [56] | 44.16 | 45.71 | 36.79 | 53.57 | 37.50 | 45.45 | 43.98 |
| GPT-4V [55]+SoM [57] | 75.00 | 55.23 | 42.45 | 54.46 | 49.03 | 47.27 | 54.33 |
| LLaVA-v1.6-34B [56]+SoM [57] | 44.16 | 40.01 | 33.96 | 47.32 | 41.34 | 46.36 | 42.31 |
| KOSMOS-2 [8] | 28.33 | 15.23 | 4.71 | 26.78 | 12.50 | 12.72 | 17.04 |
| RegionVILA-7B [19] | 30.83 | 47.61 | 35.84 | 44.64 | 35.57 | 49.09 | 40.48 |
| SpatialRGPT-7B$_{(rgb)}$ | **99.17** | 99.04 | 79.24 | 89.28 | 83.65 | 87.27 | 89.80 |
| SpatialRGPT-7B | **99.17** | 99.04 | 80.19 | 91.96 | 87.50 | 91.81 | 91.78 |
| SpatialRGPT-VILA-1.5-3B | **99.17** | **100.0** | 81.13 | 88.39 | 85.57 | **93.63** | 91.47 |
| SpatialRGPT-VILA-1.5-8B | **99.17** | **100.0** | **84.90** | 89.28 | **91.34** | 90.90 | **92.69** |

| | Direct Distance | Horizontal Distance | Vertical Distance | Width | Height | Direction |
|---|---|---|---|---|---|---|
| GPT-4 [55] | 21.6 / 1.29 | 11.5 / 2.08 | 33.0 / 0.65 | **52.3** / 0.52 | 48.1 / 1.40 | 34.6 / 83.7° |
| GPT-4V [55] | 29.7 / 0.92 | 25.4 / 2.75 | 33.0 / 0.48 | 51.1 / 0.37 | **68.4** / 1.57 | 43.9 / 69.9° |
| LLaVA-v1.6-34B [56] | 24.3 / 0.76 | 24.5 / 1.59 | 30.1 / 0.62 | 30.8 / 0.40 | 42.8 / 1.96 | 33.6 / 78.2° |
| GPT-4V [55]+SoM [57] | 25.7 / 1.02 | 22.1 / 2.36 | 33.9 / 0.64 | 45.8 / 0.70 | 62.4 / 1.08 | 54.2 / 55.5° |
| LLaVA-v1.6-34B [56]+SoM [57] | 12.8 / 1.15 | 20.4 / 1.79 | 11.3 / 0.95 | 9.0 / 0.91 | 7.5 / 3.11 | 12.8 / 33.3° |
| KOSMOS-2 [8] | 4.1 / >10 | 4.91 / >10 | 1.9 / 2.26 | 3.0 / 5.42 | 1.5 / 3.82 | 1.9 / 104° |
| RegionVILA-7B [19] | 22.3 / 1.30 | 24.6 / 3.26 | 17.9 / >10 | 36.8 / >10 | 49.6 / 1.61 | 35.5 / 79.8° |
| SpatialRGPT-7B$_{(rgb)}$ | 35.1 / 0.35 | 59.0 / 0.27 | 53.8 / **0.27** | 51.9 / 0.31 | 54.9 / 0.63 | 95.3 / 17.1° |
| SpatialRGPT-7B | 41.2 / 0.33 | 65.6 / 0.25 | **51.9** / **0.27** | 49.6 / 0.31 | 57.9 / 0.61 | 95.3 / 15.4° |
| SpatialRGPT-VILA-1.5-3B | 44.6 / **0.30** | 63.1 / 0.22 | 50.9 / 0.28 | 42.9 / 0.33 | 63.2 / 0.60 | 93.5 / 10.4° |
| SpatialRGPT-VILA-1.5-8B | **45.9** / 0.31 | **68.0** / 0.22 | 56.6 / 0.28 | 48.9 / **0.28** | 61.7 / **0.41** | **95.3 / 9.7°** |

Table 1: SpatialRGPT-Bench results. ☐ are Blind LLMs with Language Referral. ☐ are VLMs with Language Referral. ☐ are Region-aware VLMs. Numbers in the top table represent success rates (↑), while the bottom table includes success rates (↑) and absolute relative error (↓).

| | $VQA_{v2}$ | GQA | $SQA^I$ | $VQA^T$ | POPE | MME | MMB |
|---|---|---|---|---|---|---|---|
| VILA-1.5-3B | 80.4 | 61.5 | 69.0 | 60.4 | **85.9** | **1442** | 63.4 |
| SpatialRGPT-VILA-1.5-3B | **81.1** | **62.3** | **71.0** | **61.7** | 85.5 | 1424 | **65.6** |

| | MMB-CN | SEED | $SEED^I$ | $MMMU_V$ | $MMMU_T$ | $LLaVA^B$ | MMVet |
|---|---|---|---|---|---|---|---|
| VILA-1.5-3B | 52.7 | 60.9 | 67.9 | **33.3** | 30.8 | **75.9** | 35.4 |
| SpatialRGPT-VILA-1.5-3B | **53.6** | **61.8** | **69.0** | 33.0 | **31.3** | 71.5 | **38.2** |

Table 2: Comparison of SpatialRGPT and base model performance on general VLM benchmarks.

## 4.1   3D Spatial Reasoning Benchmarks

Currently, there are no visual-language benchmarks that specifically focus on VLM's ability to understand 3D spatial concepts like metric distance or size differences between objects. Recently, SpatialVLM created a spatial reasoning VQA benchmark using human labelers to annotate spatial information on 2D images, but this benchmark is not publicly available. To address this gap, we develop SpatialRGPT-Bench, a spatial reasoning VQA benchmark using data from both urban (nuScenes [58], KITTI [59]) and indoor (SUNRGBD [60], ARKitScenes [61]) environments, as well as simulated scenes (Hypersim [62]). These datasets cover various potential applications and include diverse object types, enhancing our benchmark's thoroughness. We use preprocessed 3D cuboids for each object from Omni3D [63], all positioned within a unified 3D camera coordinate system and categorized by object classes. With these 3D cuboid annotations, we developed a conversational benchmark using our data generation pipeline. This benchmark comprises 657 qualitative and 749 quantitative VQA pairs, covering 88 distinct classes. All the samples come from the validation or test splits of the original datasets and are unseen by SpatialRGPT during the training phase. Please see Appx. E for statistics and examples of SpatialRGPT-Bench.

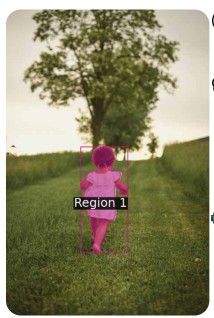

Q: Can you estimate the age of Region [1]? Think step by step.

GPT4-V
In Region [1], the individual appears to be a toddler, possibly around 1 to 3 years old, judging from their size, posture, and the style of walking, which suggests early childhood.

The height of Region [1] is 3.77 feet. Since the average height of a 3-year-old is around 3.5 feet, so Region [1] is likely a 3-year-old.

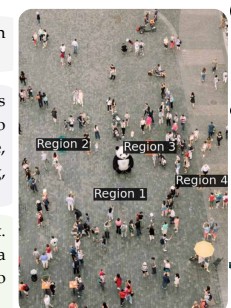

Q: If you were the person at Region [1], consider Region [2], Region [3], and Region [4], which one is standing at your 10'oclock direction?

GPT4-V
For the person in Region [1], Region [3] is in the 10 o'clock direction. This region is to the upper left from their perspective, aligning with the general direction indicated by the 10 o'clock position on a clock.

The person in Region [1] will see the person in Region [2] at 10 o clock.

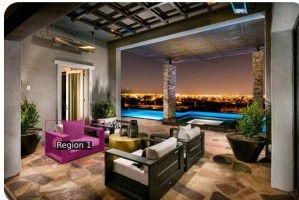

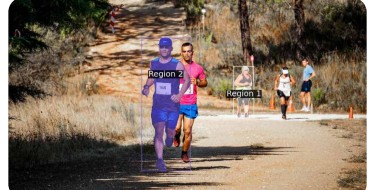

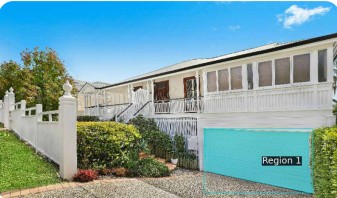

Q: How many people do you think can sit on Region [1]? Explain why.

GPT4-V
Region [1] highlights a purple sofa with a vibrant pattern, capable of comfortably seating three people. This estimation is based on the standard dimensions of sofas that usually accommodate three adults when the size and arrangement resemble what is shown in the image.

The width of Region [1] is 1.2 meters. Since the average human width is around 0.5 meters, two people can sit on Region [1].

Q: Can you estimate how much time it will take for the person in Region [1] to reach the location where Region [2] is now? Say if the person is Usain Bolt.

GPT4-V
Considering that the person is Usain Bolt, we would need some additional information such as the distance between the two regions and the average speed Usain Bolt could maintain over that distance in a running setting.

The distance between Region [1] and Region [2] is 10.84 meters. If Usain Bolt can run at a speed of 10 meters per second, it will take him 1.084 seconds to reach the location where Region [2] is now.

Q: Based on the spatial dimension of Region [1], what kind of vehicles would not fit in?

GPT4-V
In the image you've shown, Region [1] covers the garage area. To determine what type of vehicles might not fit in this garage based on its spatial dimensions, we need to estimate the size of the garage. From the image, the garage appears to be a typical residential garage. Standard single-car garages usually have a width of about 10 to 12 feet and a depth of about 20 to 24 feet.

The Region [1] is 2.7 meters in height, so vehicles taller than 2.7 meters would not fit in.

Figure 4: SpatialRGPT is capable of complex spatial reasoning, addressing gaps that current leading vision language models, such as GPT-4V, struggle with.

We consider three categories of models as baselines:

**Blind LLMs w/ Language Referral.** The blind [10] LLM model relies solely on text and generates answers using only the content of the question. To enhance this approach, we prepend the object class to each question. This method serves as a baseline to gauge how much spatial reasoning can be derived from purely existing world knowledge. We choose GPT-4 to represent this baseline, as it is the most advanced model for encapsulating comprehensive world knowledge.

**VLMs w/ Language Referral.** The setup is similar to the blind LLMs but includes access to visual content, which could allow the model to answer better than a blind LLM. We employ current state-of-the-art VLMs, GPT-4V and LLaVA-v1.6-34B [56], as baselines for this category.

**Region-aware VLMs.** This category explores models with region-level capabilities similar to our method. The models do not receive any language captions or object class information related to the region of interest; they rely solely on their visual processing capabilities. We equip GPT-4V [55] and LLaVA-v1.6-34B with Set of Marks (SoM) [57] to enable region-referring capabilities. Additionally, we include KOSMOS-2 [24], a VLM capable of taking bounding box inputs to reference objects, and RegionVILA (RegionGPT [19] with VILA [50] pre-training). RegionVILA-7B also serves as an ablation baseline to our method; it shares the same model architecture as our SpatialRGPT-7B$_{(rgb)}$ variant but is trained without our specialized spatial VQA dataset.

We use GPT-4 to evaluate the response for each model; please see Appx. J for details. For qualitative QAs, GPT-4 scores the alignment between the model's response and the correct answer as 0 or 1. For quantitative QAs, GPT-4 standardizes numerical values across units into meters; we then calculate accuracy and error metrics. We present the results in Table 1. The upper rows of the table show accuracy (correct vs incorrect or failed to answer) for qualitative QAs. The lower rows report on

| Model | mAP (↑) | Acc. (%) |
|---|---|---|
| CLIP [64] | 58.9 | - |
| RegionCLIP [65] | 58.3 | - |
| LLaVA-7B [29] | - | 40.0 |
| Shikra-7B [25] | - | 53.9 |
| GPT4RoI-7B [31] | - | 64.0 |
| PVIT-7B [66] | - | 64.5 |
| ASM-7B [32] | 69.3 | - |
| RegionGPT-7B [19] | 70.0 | 80.6 |
| SpatialRGPT-7B | 69.7 | 79.9 |
| SpatialRGPT-VILA-1.5-3B | 72.5 | 82.5 |
| SpatialRGPT-VILA-1.5-8B | **72.9** | **82.9** |

Table 3: Region-level classification results. We follow the evaluation in RegionCLIP [65] and RegionGPT [19], report the results of object classification with ground-truth box on COCO-2017 validation set.

| Model | Acc. (%) |
|---|---|
| Qwen-VL-Max [67] | 58.9 |
| Gemini Pro [68] | 50.0 |
| Claude 3 OPUS [69] | 57.3 |
| GPT-4V-*preview* [55] | 58.9 |
| GPT-4V-*Turbo* [55] | 66.9 |
| GPT-4o [55] | 64.5 |
| InstructBLIP-13B [51] | 50.0 |
| Yi-VL-34B [70] | 53.2 |
| LLaVA-v1.5-13B-xtuner [71] | 54.0 |
| LLaVA-v1.6-34B [56] | 64.5 |
| MiniGPT-4-v2-7B [26] | 49.2 |
| InstructBLIP-7B [51] | 50.8 |
| LLaVA-v1.5-7B-xtuner [71] | 50.8 |
| CogVLM-7B [27] | 50.8 |
| LLaVA-v1.5-7B [72] | 51.6 |
| LLaVA-InternLM2-7B [73] | 52.4 |
| SpatialRGPT-7B | 82.3 |
| SpatialRGPT-VILA-1.5-8B | **87.9** |

Table 4: BLINK$_{RelativeDepth}$ results.

quantitative QAs, detailing their success rate (answers within $\pm 25\%$ of the ground truth value) and the absolute relative error [43, 42]. We exclude answers that failed to produce a numerical response from the relative error calculations. The results show that SpatialRGPT significantly outperforms baselines in terms of success rate for qualitative QAs and maintains the lowest error rate for quantitative QAs. Interestingly, we found that blind LLMs and VLMs with language referrals achieved commendable success rates for quantitative QAs, especially for questions related to width and height. This suggests that LLMs can accurately answer specific spatial questions using their extensive world knowledge. Additionally, our SpatialRGPT-7B variant demonstrates improved performance over the SpatialRGPT-7B$_{(rgb)}$ variant, especially in scenarios where relative depth information can be used to resolve ambiguities, such as distinguishing between behind/front, wide/thin, and estimating distances.

## 4.2 Public Vision-language Benchmarks

**General Benchmarks.** In this section, we evaluate whether integrating spatial VQA data and depth information affects performance on other VQA tasks. We compared our models with VILA-1.5-3B, which is trained on the same general VQA datasets. As shown in Table 2, our variants performed similarly to the baselines and slightly better on the VQA-v2 and MMVet datasets. These results align with findings from [17], indicating that VLMs generally underperform on spatial reasoning tasks but can improve with specific spatial VQA training without compromising general VQA performance.

**Region & Spatial Benchmarks.** We follow the evaluation protocol from RegionGPT [19] and report object classification results using ground-truth boxes on the COCO-2017 validation set. As shown in Table 3, SpatialRGPT outperforms the baselines, demonstrating its strong region cognition capabilities. We further evaluate SpatialRGPT on BLINK [9]'s Relative Depth Benchmark. This benchmark is particularly challenging as it assesses point-level depths, while both the point-level region input and point-level questions were not specifically included in the training of SpatialRGPT. We use bounding boxes to mark the target points and evaluate the test set online with the EvalAI server. As shown in Table 4, SpatialRGPT significantly outperforms the state-of-the-art, achieving over 20% accuracy gain compared to GPT-4V-Turbo. Our model demonstrated strong performance, highlighting its ability to generalize to new tasks without explicit training.

## 4.3 Real-world Applications

**Complex Spatial Reasoning.** In this application, we aim to explore whether SpatialRGPT can function as a complex spatial reasoner on its own. Unlike the system mentioned in [17], which uses GPT-4 to handle reasoning tasks and employs VLM solely for answering basic spatial queries, SpatialRGPT directly integrates these capabilities. We provide examples in Figure 4, where we compare SpatialRGPT's responses to those from GPT-4V using real-world samples. Our model demonstrates the ability to address complex spatial questions based on its own spatial knowledge. This suggests that SpatialRGPT has developed a robust representation of spatial learning and that this knowledge has effectively generalized to enhance its intrinsic language reasoning abilities.

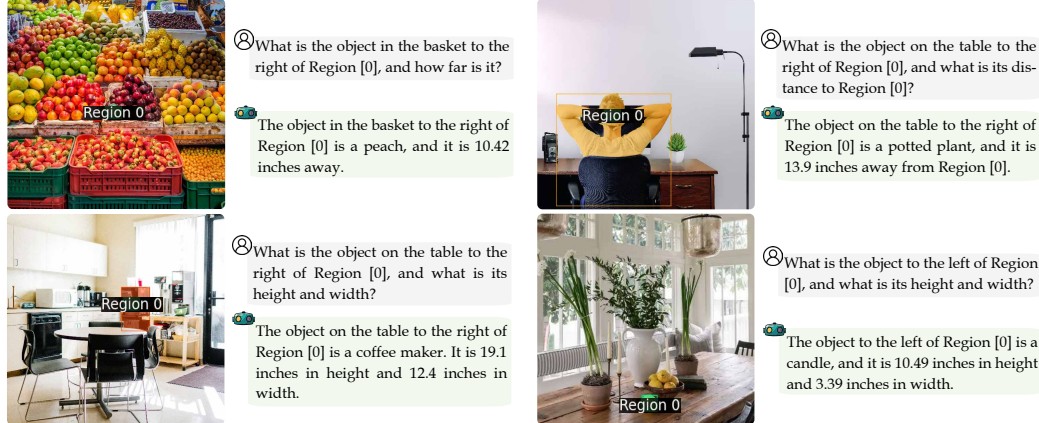

Figure 5: Examples of SpatialRGPT performing multi-hop reasoning.

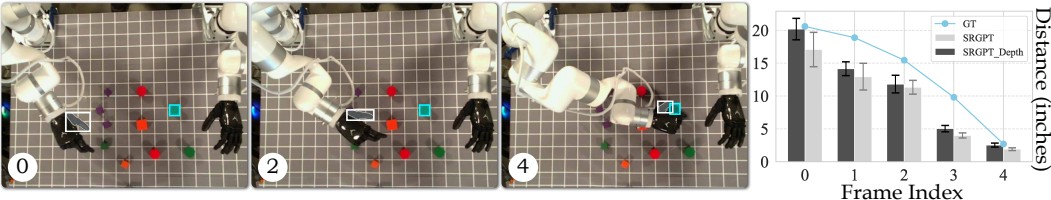

Figure 6: SpatialRGPT functions as a region-aware reward annotator. The estimated distance decreased monotonically as the fingertip moves towards the target.

**Multi-hop Reasoning.** In Figure 5, we show examples of SpatialRGPT handling multi-hop reasoning. In the upper left sample, the model first identifies what's to the right of Region [0] (a single apple), finds the basket there, determines what's inside the basket, and then provides spatial details about the object inside. Even though our training data doesn't specifically include such multi-hop tasks, SpatialRGPT can still manage them effectively. This indicates that the model has developed a strong understanding of spatial relationships.

**Region-aware Dense Reward Annotator.** Recently, [17] has shown that VLMs can function as dense reward annotators for robotics tasks by specifying tasks in natural language and having the model annotate rewards for each frame in a trajectory. However, this approach can be constrained by the language's ambiguity, especially when multiple identical objects are present or when targeting a small, specific region in a scene, which can be difficult to describe precisely with language alone. Given that SpatialRGPT is equipped with region-aware capabilities, we can directly specify the regions of interest. To study this application, we conducted a real robot experiment. Specifically, we defined two regions using bounding boxes (one for the fingertip and one for a green cube) and tasked SpatialRGPT to annotate rewards using the distance between the two regions. The results, shown in Figure 6, indicate that the estimated distance between the fingertip and its target cube decreased monotonically as the fingertip moved towards its goal. Also, our depth variant performs slightly better than the RGB variant. This demonstrates SpatialRGPT 's effectiveness as a region-aware dense reward annotator, offering a more precise and efficient alternative to language-only approaches.

## 5 Discussion

**Conclusion.** We introduce SpatialRGPT, a novel framework designed to enhance the spatial reasoning capabilities of Vision Language Models (VLMs). By integrating a region representation module and a flexible plugin for depth information, SpatialRGPT allows VLMs to effectively perceive spatial arrangement at both local and global scopes. Our data curation pipeline facilitates the learning of 3D spatial knowledge from scene graphs, while SpatialRGPT-Bench provides a comprehensive benchmark for evaluating spatial cognition across diverse environments. The results demonstrate significant improvements in spatial reasoning tasks while showcasing the model's ability to reason complex spatial relations and perform as dense reward annotators for robotic applications.

**Limitations.** One limitation of our work is the use of Axis-Aligned Bounding Boxes (AABBs), which can result in inaccuracies in label representation. A more accurate alternative is oriented bounding boxes (OBBs), but implementing them requires precise object pose estimation, which remains challenging due to the lack of open-world solutions. The most accurate approach would be human labeling [74], while this requires significant effort. We leave these for future work.

**Acknowledgement.** This work was supported, in part, by the Qualcomm Innovation Fellowship.

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

# Appendix Table of Contents

# A    Ablation Study on Augmented SpatialRGPT-Bench

We conduct additional experiments by augmenting and rephrasing both questions and answers in SpatialRGPT-Bench using GPT-4. The results are shown in Table 5. The results show that SpatialRGPT consistently outperforms the baseline models, even when the questions and answers are different from the training data.

| | Below/ Above | Left/ Right | Big/ Small | Tall/ Short | Wide/ Thin | Behind/ Front | Qualitative Average |
|---|---|---|---|---|---|---|---|
| GPT-4V-*Turbo* | 66.7 | 47.6 | 66.0 | 64.2 | 71.1 | 47.2 | 60.5 |
| SpatialRGPT-7B | **95.8** | **99.0** | **77.4** | **92.9** | **82.7** | **90.9** | **90.0** |

| | Direct Distance | Horizontal Distance | Vertical Distance | Width | Height | Direction |
|---|---|---|---|---|---|---|
| GPT-4V-*Turbo* | 30.4 / 0.87 | 26.2 / 2.66 | 33.9 / 0.51 | 48.8 / 0.35 | **69.1** / 1.35 | 40.1 / 70.0° |
| SpatialRGPT-7B | **43.2 / 0.32** | **63.9 / 0.27** | **52.8 / 0.26** | **51.1 / 0.31** | 54.1 / **1.02** | **95.3 / 15.3°** |

Table 5: Augmented SpatialRGPT-Bench results. Numbers represent success rates (↑) and absolute relative error (↓).

# B    Ablation Study on Metric-Scale Width and Height Data

We conduct an ablation study to see if adding width and height data affects other types of questions. As shown in Table 6, adding this data slightly improved the accuracy for questions about size (like big/small, tall/short, wide/thin) but slightly worsened the accuracy for questions about the distance between objects (horizontal and vertical). This suggests that information about object size helps with size-related questions but might make distance measurements less clear.

| | Below / Above | Left / Right | Big / Small | Tall / Short | Wide / Thin | Behind / Front | Avg. |
|---|---|---|---|---|---|---|---|
| − width & height | 99.1 | 99.0 | 75.8 | 90.8 | 82.8 | 92.1 | 90.5 |
| + width & height | 99.1 +0 | 99.0 +0 | 80.1 +4.3 | 91.9 +1.1 | 87.5 +4.7 | 91.8 -0.3 | 90.5 +1.2 |

| | Direct Distance | Horizontal Distance | Vertical Distance | Width | Height | Direction |
|---|---|---|---|---|---|---|
| − width & height | 41.2 | 69.3 | 54.8 | 22.8 | 21.2 | 95.1 |
| + width & height | 41.2 +0 | 65.6 -3.7 | 51.9 -2.9 | 49.6 +26.8 | 57.9 +36.7 | 95.3 +0.2 |

Table 6: Ablation study on the impact of width and height data on the performance of other categories. Numbers represent success rates (↑).

# C    Ablation Study on Bounding Box Types

We conduct an ablation study to examine the effect of using axis-aligned bounding boxes (AABB) versus PCA-based oriented bounding boxes (OBB). For this study, we use human-labeled OBBs from the Omni3D test set as the ground truth. We then compare the mean-square error of the width and height measurements for AABBs and PCA-based OBBs labeled by our 3D scene graph pipeline. The results are shown in Table 7. PCA-based OBB often lacks accuracy due to the incomplete and noisy nature of point clouds captured from a single view.

| BBox Type | Width (↓) | Height (↓) |
|---|---|---|
| Oriented BBox | 17.09 | 4.83 |
| Axis-aligned BBox | 8.27 | 2.35 |

Table 7: Ablation study on axis-aligned vs. oriented bounding boxes. Numbers indicate MSE comparing to Omni3D ground truth.

# D  Ablation Study on Different Input Modalities

As mentioned in Section 3.4, SpatialRGPT can take both boxes and masks as input during the inference phase. In this study, we aimed to test the impact of box and mask inputs on our SpatialRGPT-Bench. We presented the results in Table 8, where we observed a slight drop in performance when using boxes, but in general, the performance was very close. This suggests that the random modality strategy used during training is effective.

| | Below/ Above | Left/ Right | Big/ Small | Tall/ Short | Wide/ Thin | Behind/ Front | Avg. |
|---|---|---|---|---|---|---|---|
| SpatialRGPT-7B-Mask | 99.17 | 99.04 | 80.19 | 91.96 | 87.50 | 91.81 | 91.78 |
| SpatialRGPT-7B-Box | 99.17 | 98.09 | 83.01 | 91.96 | 82.69 | 92.72 | 91.47 |

| | Direct Distance | Horizontal Distance | Vertical Distance | Width | Height | Direction |
|---|---|---|---|---|---|---|
| SpatialRGPT-7B-Mask | 41.2 / 0.33 | 65.6 / 0.25 | 51.9 / 0.27 | 49.6 / 0.31 | 57.9 / 0.61 | 95.3 / 15.4° |
| SpatialRGPT-7B-Box | 39.2 / 0.35 | 63.1 / 0.25 | 56.6 / 0.27 | 48.8 / 0.36 | 60.1 / 1.06 | 94.3 / 10.2° |

Table 8: Ablation study on effect of different input modalities to Spatial RGPT. Numbers in the top table represent success rates (↑), while the bottom table includes success rates (↑) and absolute relative error (↓).

# E  Statistics and Samples of SpatialRGPT-Bench

Figure 7 presents key statistics from our SpatialRGPT-Bench, including counts for QA categories, data sources, and objects. We categorize the QA data into 12 distinct types, evenly divided between relative relationships and metric measurements. Notably, some datasets, such as SUNRGBD, emphasize close-object scenarios. To reduce bias, we source our data from a diverse range of datasets following [63]. We also show six samples from our SpatialRGPT-Bench in Figure 8.

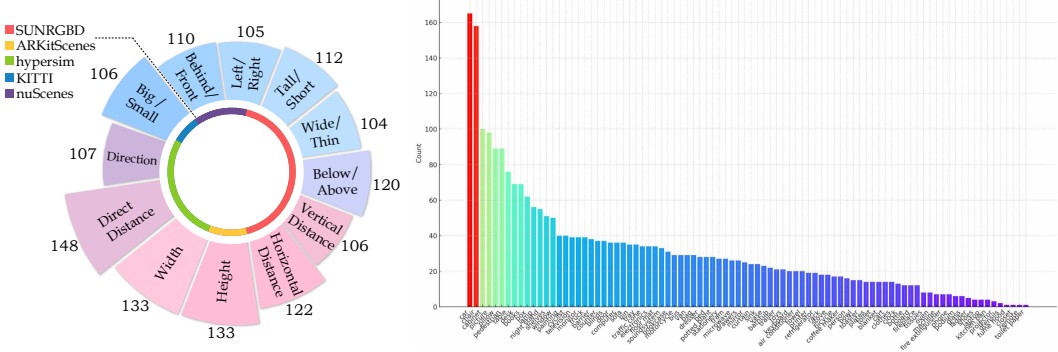

Figure 7: SpatialRGPT-Bench statistics. Left: Category count and source count. Right: Object count.

# F  Implementation Details for Data Pipeline

In this section, we aim to provide a detailed implementation of our data annotation pipeline and intermediate results obtained through each component.

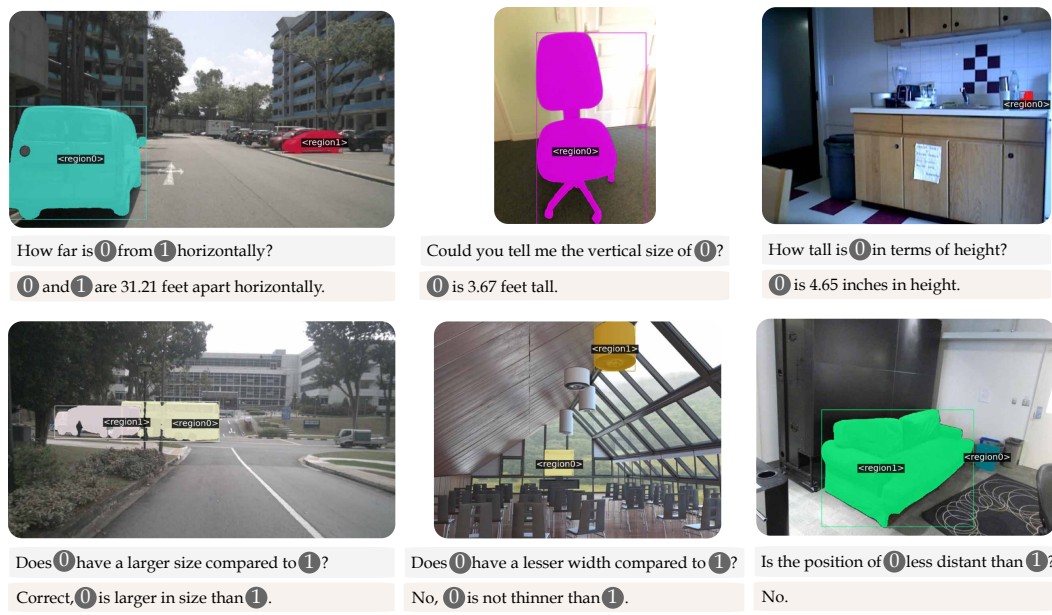

Figure 8: Samples in SpatialRGPT-Bench.

## F.1  Filtering.

Recent VLMs often benefit from the broad capabilities gained through training with large-scale 2D image datasets [75, 49]. However, many images in these datasets are unsuitable for developing spatial reasoning QA. For instance, some images may be computer screenshots, paintings, collages, or simply a piece of text. Similar to SpatialVLM [17], we use a CLIP-based open-vocabulary classification model [76] to identify and exclude these unsuitable images. We follow the labeling used in SpatialVLM but have made a few adaptations to better suit the data distribution of the OpenImage [49] dataset. We show the labels we use in Listing 1. With this process, we filtered out 700K samples from the 1.7M OpenImage samples.

Listing 1: CLIP labels used during filtering.

```
positive_labels = [
    "a DSLR photo of an indoor scene",
    "a DSLR of an outdoor scene",
    "an iphone photo of an indoor scene",
    "an iphone photo of an outdoor scene",
]

negative_labels = [
    "a close up shot of a single object",
    "a product displayed in front of a white back ground",
    "a painting",
    "a collage of images",
    "a screenshot of graphics user interface",
    "a piece of text"
]
```

## F.2  Metric Depth Estimation

As stated in the main paper, we choose Metric3Dv2 as our metric depth estimator. We have observed that Metric3Dv2 and WildCamera's camera intrinsic perform well on images taken in natural environments. In this section, we present the predicted normal maps from the depth model on OpenImages. These normal maps can be viewed as a proxy to estimate the quality of the reconstructed geometry's edges.

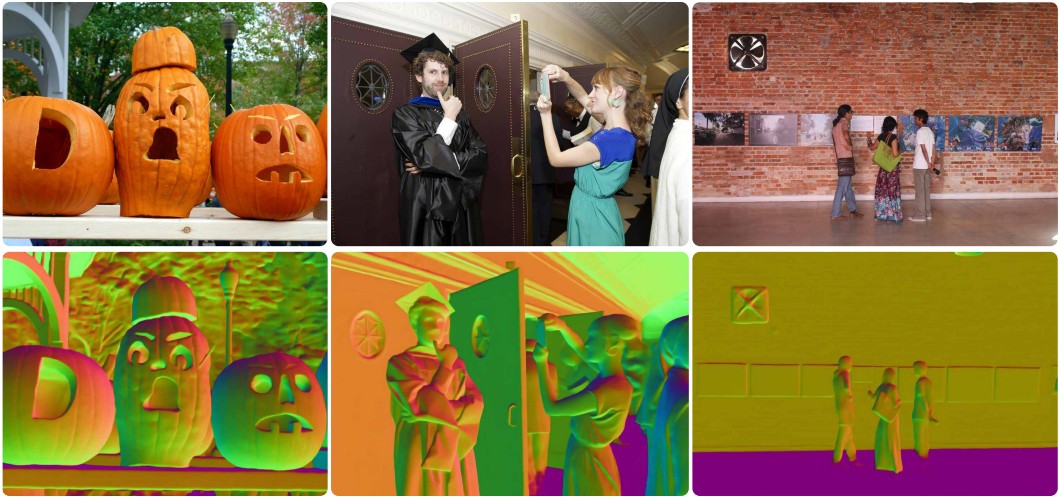

Figure 9: Predicted normal maps using Metric3Dv2 and WildCamera.

## F.3 Point Cloud Processing

Here, we detailed how we process the point clouds into scene graphs.

**Canonicalization.** Our canonicalization method is straightforward. After obtaining the pitch and roll through PerspectiveFields, we transform the point cloud into a canonicalized space using the inverse of the rotation matrix. Figure 10 illustrates the successful alignment of the ground surface with the z-axis angle after canonicalization. This process ensures that the axis-aligned bounding box accurately represents the vertical information of the objects, such as height and vertical distance. Our simple yet effective approach liberates our method from surface segmentation and RANSAC. We have empirically found this procedure robust for most natural images taken by cameras in real-world conditions.

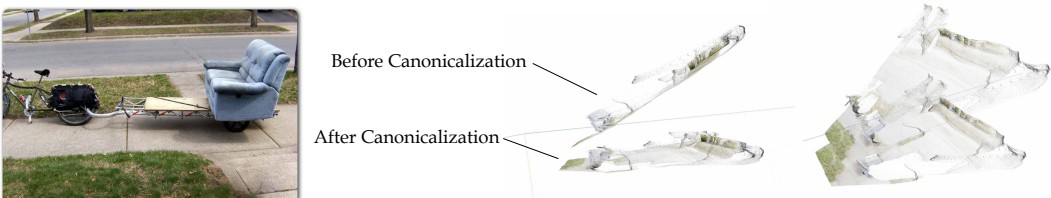

Figure 10: Canonicalization Results.

**Denoising and constructing axis-aligned bounding box.** The point clouds obtained from single-view depth may contain noise. Following [17, 22], we carry out several denoising steps based on the approach to filter out outliers and unwanted points, thereby improving the robustness and accuracy of the bounding box. Initially, we eliminate statistical outliers from the object points and then downsample the data to a lower resolution. Subsequently, we use DBSCAN to further remove noise. If the points of an object are fewer than ten after DBSCAN clustering, we exclude that object area. Finally, we employ Open3D to create axis-aligned bounding boxes for each object. The pseudocode for our denoising process is as in Listing 2.

Listing 2: Point cloud denoising steps.

```
def process_pcd(pcd):
    scale = norm(pcd).std * 3.0 + 1e-6
    [pcd, _] = pcd.remove_statistical_outlier(nb_neighbors=10, std_ratio
        =1.2)
    pcd = pcd.voxel_down_sample(voxel_size=max(0.01, scale/40))
    pcd = pcd_denoise_dbscan(
        pcd, eps=0.2, min_points=10
    )
    return pcd
]
```

## F.4 Open Spatial Dataset QA Templates

We provide samples for each category of QA in the templates that we use to generate QAs mentioned in Section 3.1.

Listing 3: Template for QA synthesis.

```
distance_template_questions = [
   "What is the distance between [A] and [B]?",
   "How far away is [A] from [B]?",
   "Can you provide the distance measurement between [A] and [B]?",
]
distance_template_answers = [
   "[A] and [B] are [X] apart.",
   "A distance of [X] exists between [A] and [B].",
   "[A] and [B] are [X] apart from each other.",
]
left_predicate_questions = [
   "Is [A] to the left of [B] from the viewer's perspective?",
   "Does [A] appear on the left side of [B]?",
   "Can you confirm if [A] is positioned to the left of [B]?",
]
left_true_responses = [
   "Yes, [A] is to the left of [B].",
   "Indeed, [A] is positioned on the left side of [B].",
   "Correct, you'll find [A] to the left of [B].",
]
left_false_responses = [
   "No, [A] is not to the left of [B].",
   "In fact, [A] is to the right of [B].",
   "Incorrect, [A] is not on the left side of [B].",
]
direction_questions = [
   "If you are at [A], where will you find [B]?"
]
direction_responses = [
   "[B] is roughly at [X] o'clock from [A].",
   "[A] will find [B] around the [X] o'clock direction."
]
```

## F.5 LLM Prompts for Complex QA

```
messages = [ {"role":"system", "content": f""" You are a helpful assistant tasked
with generating spatial reasoning-based questions and answers from provided descriptions of scenes.
Always craft a question without directly revealing specific details from the description. Always
generate questions related to the description using <regionX>. The description should always be used
to answer and not leak into the question. When mentioning the objects or regions, use <regionX>
instead of the objects or regions. Speak like you are the observer's perspective. Always make sure all
the description objects or regions are mentioned with <regionX> in the question. """}
]
for sample in fewshot_samples:
   messages.append({"role":"user", "content":sample['context']})
   messages.append({"role":"assistant", "content":sample['response']}
   )
messages.append({"role":"user", "content":'\n'.join(query)})
```

Table 9: Llama-3 prompts for complex QA synthesis.

# G   Implementation Details for SpatialRGPT Architecture

## G.1   Visual Backbone.

For SpatialRGPT-7B, we adopt a pre-trained OpenAI CLIP-L model [64] as the visual backbone. As for SpatialRGPT-VILA-1.5-3B/8B, we use 384×384 image resolutions with SigLIP [77] to include more visual details for the model, which can help with vision language tasks that require fine-grained details [50] and are beneficial for region-level representations [34].

## G.2   Region-feature Extractor.

We adopt the region feature extraction technique in [19]. To begin with, we use a feature refinement module consisting of a 2-layer deconvolution network designed to upscale the original feature map. Then, we employ MaskPooling to extract and average the refined features from the masked area.

## G.3   Multi-modal Connector

To bridge representations from various modalities (e.g., image to language, depth to language), we employ a simple linear layer. Following the approach suggested in [50], using a straightforward connector helps the LLM to concentrate more on processing visual inputs, thereby enhancing generalization. We implement two separate connectors, one for image embeddings and another for depth embeddings, to ensure that each modality is handled distinctly. This separation prevents the mixing of modalities, which could otherwise compromise the effectiveness of the model. Note that for SpatialRGPT-VILA-1.5-3B/8B, we follow VILA-1.5 and use a two-layer MLP as our connector.

# H   Implementation Details for Training SpatialRGPT

## H.1   Instruction Tuning Data

Here, we list the instruction tuning data we use in addition to the OSD dataset. Includes general instruction tuning datasets from LLAVA-1.5 [29], LAN-style instructions from VILA [50] (listed in Table 10) and the region-level instruction tuning data from [19] (listed in Table 11) that we use in stage three of the training.

| Categories | Datasets |
|---|---|
| Captioning | Image Paragraph Captioning [78], MSR-VTT [79], TextCaps [80] |
| Reasoning | CLEVR [81], NLVR [82], VisualMRC [83] |
| Translation | Multi30k [84] |
| VQA | ActivityNet-QA [85], DocVQA [86], GQA [87], iVQA [88], MSRVTT-QA [79], MSVD-QA [79], OCR-VQA [89], ST-VQA [90], ViQuAE [91], VQAv2 [92], Visual Dialog [93] |

Table 10: The general SFT blend [50] we used.

| Categories | Datasets |
|---|---|
| Classification | V3Det [94], COCO [40], LVIS [95] |
| Caption | V3Det [94] VG [96], RefCOCO [97] |
| Relationship | VG [96] |
| REC | RefCOCO [97] |

Table 11: The region-level SFT blend [19] we used.

## H.2   Hyperparameters

Please refer to VILA's paper on the implementation of the hyperparameters used in the first two stages. In the instruction fine-tuning stage, the maximum learning rate is reduced to 2e-5, and the batch size is adjusted to 16. All other hyperparameters remain the same as in the pre-training stage.

# I  Experimental Setting and Details

## I.1  Experiments Compute Resources

**Open Spatial Dataset.**  Our Open Spatial Dataset uses images from OpenImages, which contains a total of 1.7 million images. Our data preprocessing pipeline was tested on a system with 8 GPUs. The filtering process for 1.7 million images takes 4 hours and results in 1 million samples. The camera calibration and metric depth estimation each took around 4 hours. Note that the depth estimation requires our estimated camera intrinsics as input, so these two processes cannot be parallelized. The open-vocabulary detection and segmentation process takes 8 hours. As the process involves sequential operations, we did not specifically optimize it for parallelization. For LLM-based QA synthesis, we employ LLama3-70b using sglang backend, which takes 12 hours. In general, the total time required to convert OpenImages into 3D scene graphs is within a day, and constructing the QAs takes another half.

**SpatialRGPT Training.**  The first two stages of Spatial RGPT are inherited from VILA [50], which is trained on 16 A100 GPU nodes, with each node having 8 GPUs. The training times for each stage of the 7B model are as follows: connector initialization takes 4 hours, visual language pre-training takes 30 hours. The depth connector is further pre-trained using 2 A100 GPU nodes, taking 4 hours. The final visual instruction-tuning is also experimented on 2 A100 GPU nodes, taking 12 hours.

**SpatialRGPT-Bench.**  The SpatialRGPT-Bench dataset is created from ground truth 3D cuboids and human-annotated labels. Masks only need to be generated when bounding boxes are provided. We use SAM-HQ in our data pipeline to convert the bounding boxes into masks, which takes approximately 4 hours to process 10,000 samples. After this, we synthesize QA and randomly select 1,500 samples. Subsequently, we conduct human verification to filter out incorrect annotations, which takes a day to complete.

# J  Benchmark Evaluation Details

Our benchmark poses a challenge in evaluation due to the possibility of multiple correct answers in different units. Typically, human trials, like those used by [17], could handle this but are often too slow and costly, mainly as our benchmarks include over a thousand samples. As an alternative, we employ GPT-4 [55] to assess correctness. The evaluation process involves providing a question, the correct answer, and the model's response to the LLM. For qualitative questions, GPT-4 determines if the model's response aligns with the correct answer by assigning a score of 0 or 1. For quantitative questions, GPT-4 extracts numerical values from both the correct answer and the model's response, converting them to the same unit (such as meters). We then measure the accuracy and error of the model's response based on this standardized unit. We provide prompts we use in Table 13 and Table 12.

```
messages = [{"role":"system", "content": f"""You are a helpful assistant designed
to output JSON.

You should help me to evaluate the response given the question and the correct answer.
To mark a response, you should output a single integer between 0 and 1.

(1) means that the response perfectly matches the answer.
(0) means that the response is completely different from the answer."""}
]
for sample in fewshot_samples:
    messages.append({"role":"user", "content":sample['context']})
    messages.append({"role":"assistant", "content":sample['response']}
    )
messages.append({"role":"user", "content":'\n'.join(query)})
```

Table 12: GPT-4 prompts for SpatialRGPT-Bench qualitative evaluation.

```
messages = [{"role":"system", "content": f"""You are a helpful assistant designed
to output JSON.

You should help me to evaluate the response given the question and the correct answer.
You need to convert the distance of the correct answer and response to meters.
The conversion factors are as follows:
1 inch = 0.0254 meters. 1 foot = 0.3048 meters. 1 centimeter (cm) = 0.01 meters.
You should output two floats in meters, one for the answer, and one for the response."""}
]
for sample in fewshot_samples:
    messages.append({"role":"user", "content":sample['context']})
    messages.append({"role":"assistant", "content":sample['response']}
    )
messages.append({"role":"user", "content":'\n'.join(query)})
```

Table 13: GPT-4 prompts for SpatialRGPT-Bench quantitative evaluation.

# K   More Discussion on Limitations

For the most accurate object detection, oriented bounding boxes (OBB) are preferred over axis-aligned bounding boxes (AABB). As illustrated in Figure 11, the dimensions obtained from AABBs can differ from those obtained with OBBs. There are two methods to compute an OBB. A simple method involves calculating the OBB using Principal Component Analysis (PCA) of the object's convex hull, which provides an approximate minimal bounding box. However, this approximation often lacks accuracy due to the incomplete and noisy nature of point clouds captured from a single view. Furthermore, this method still cannot handle extreme cases when objects are partially elevated (see Appdx C). The most precise method involves determining the OBB based on the object's pose, which is currently challenging due to limitations in obtaining accurate object poses. Future improvements could include integrating available pose estimation approaches. However, currently, there are no open-vocabulary solutions for object pose estimation, so this remains an area for future research. Another direction, explored in subsequent work (e.g., Q-Spatial Bench [74]), addresses this limitation by leveraging human labeling.

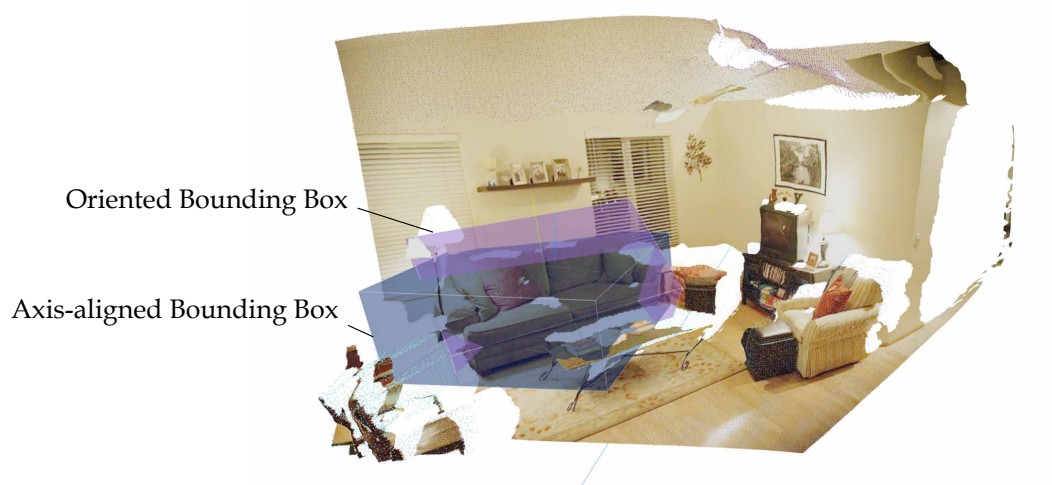

Oriented Bounding Box

Axis-aligned Bounding Box

Figure 11:  Different types of bounding box.

## L  Broader Impacts

SpatialRGPT serves as a general-purpose visual assistant, similar to other VLMs. It offers potential benefits and risks due to its integration of LLMs. SpatialRGPT shares similar concerns with LLMs, such as output hallucinations, inherited biases from base models, and energy consumption during upscaling. Evaluating SpatialRGPT's performance is also challenging, particularly in accurately measuring the spatial information. This is an area for future enhancement, especially in the field of robotics, which values safety. Despite these challenges, releasing SpatialRGPT to the research community would be beneficial, as it would foster further development and improvement of robotics applications.

## M  Licenses

1. The training data we use, OpenImages [49], is released under `Apache License 2.0`.

2. Our paper contain images from Unsplash [98], which is released under `Unsplash License`, allowing use of photos for free, including for commercial purposes, without attributing the photographer or Unsplash.

