# OpenReview forum: "SpatialRGPT: Grounded Spatial Reasoning in Vision-Language Models"
_NeurIPS.cc/2024/Conference — NeurIPS 2024 poster_

### Official Review · Reviewer_v4dm · 2024-07-11

**Soundness:** 3
**Presentation:** 4
**Contribution:** 3
**Rating:** 7
**Confidence:** 4

**Summary:**

This paper introduces SpatialRGPT, targeting at enhancing the spatial reasoning abilities of VLMs. The authors introduce a data curation pipeline along with a benchmark that facilitates the learning and evaluation of 3D spatial knowledge. Experiments show that SpatialRGPT thrives at spatial reasoning and performs comparable to SOTA on standard VQA benchmarks. The authors also showcase some real-world applications of such a model by using it for complex spatial reasoning and as a reward annotator for robot manipulation.

**Strengths:**

1. The paper introduces a data curation pipeline that facilitates spatial relation learning, along with a benchmark that's designed specifically for this important task, which will all be made available, making it easy to follow.
2. This paper introduces a plug-and-play module that process depth information for VLM. Experimental results showcase that the module is useful for cases that involves reasoning among behind/front, wide/thin, and estimating distances.
3. Results on real robot shows that the learned spatial relation representation can indeed be used for downstream tasks.

**Weaknesses:**

1. When constructing 3D scene graph, the author mentioned using 3D axis-aligned bounding boxes for objects to compute the width and height. This could lead to inaccurate measure of the object size. It'd be great if the author could show some results on how much does this affect the overall data quality, i.e. how many objects are measured inaccurately because of the AABB assumptions of the bounding boxes.
2. When discussing the potential real-world application, the author showcased SpatialRGPT can be used as a dense reward annotator. However, the annotation process still needs manually defining regions of interest for every frame in the demonstration video, hindering its use at a broader impact.
3. The width and height result in Table 1 suggests that SpatialRGPT underperforms the best model, i.e. GPT-4V by 10.5 in success rate, this is a major concern to me since GPT-4V only has commonsense knowledge on the object size and still outperforms SpatialRGPT, which is trained on object size knowledge, by a large margin. Therefore, I'm wondering if the data of object size is useful, i.e. without this portion of data, how would SpatialRGPT perform in the task of reasoning about width and height.

**Questions:**

1. Since Open Spatial Dataset plays an important role in enhancing the ability of spatial reasoning, I'm wondering if the data that targets at different aspect would affect each other, i.e. if I remove the QA pair regarding width and height, will it affect the models performance on answering big/small questions?
2. Despite the task is targeted at region-based spatial reasoning, I'm wondering how well the model would perform on VQA tasks that includes single region but multi-hop reasoning, i.e. what is the object on the table to the right of <region1>?

**Limitations:**

The authors have adequately addressed the limitations and potential negative societal impact of their work.

---

> ### Author Rebuttal · Authors · 2024-08-07
>
> **Q:** *It'd be great if the author could show some results on how much AABB assumptions affect the overall data quality, i.e. how many objects are measured inaccurately because of the AABB assumptions of the bounding boxes.*
>
> **A:** We conduct an ablation study to examine the effect of using axis-aligned bounding boxes (AABB) versus PCA-based oriented bounding boxes (OBB). For this study, we use human-labeled OBBs from the Omni3D test set as the ground truth. We then compare the mean-square error of the width and height measurements for AABBs and PCA-based OBBs labeled by our 3D scene graph pipeline. The results are shown in Table 5 (response pdf). As noted in L747 of our paper, PCA-based OBB often lacks accuracy due to the incomplete and noisy nature of point clouds captured from a single view.
>
> ---
>
> **Q:** *When SpatialRGPT functioned as a dense reward annotator, the annotation process needed to manually define regions of interest for every frame in the demonstration video, hindering SpatialRGPT’s use at a broader impact.*
>
> **A:** SpatialRGPT can be combined with video segmentation approaches such as SAM-v2. The video can be annotated by clicking on the object (point prompt) only in the first frame.
>
> ---
>
> **Q:** *The width and height result in Table 1 suggests that SpatialRGPT underperforms the best model, GPT-4V by 10.5 in success rate.*
>
> **A:** The success rates presented in Table 1 (submission) are based on a threshold cap. Despite a lower success rate, SpatialRGPT still outperforms GPT-4V in terms of absolute relative error for both width and height measurements. The lower success rate is attributed to SRGPT-Bench being derived from Omni3D and operating within a closed-set setting, with objects from a limited number of commonly seen classes (e.g., human, car, chair), which are usually easy for GPT-4 models to estimate within a reasonable range. Therefore, the advantages of SpatialRGPT are not fully apparent under these conditions. Creating an open-world or more diverse 3D ground truth annotation dataset will provide a more comprehensive evaluation framework and can better demonstrate the strengths of SpatialRGPT. We leave this as future work.
>
> ---
>
> **Q:** *Will the data that targets at different aspect would affect each other? i.e. if we remove the QA pair regarding width and height, will it affect the models performance on answering big/small questions?*
>
> **A:** Following the reviewer’s advice, we conduct a study to see if adding width and height data affects other types of questions. As shown in Table 7 (response pdf), adding this data slightly improved the accuracy for questions about size (like big/small, tall/short, wide/thin) but slightly worsened the accuracy for questions about the distance between objects (horizontal and vertical). This suggests that information about object size helps with size-related questions but might make distance measurements less clear.
>
> ---
>
> **Q:** *How well the model would perform on VQA tasks that includes single region but multi-hop reasoning, i.e. what is the object on the table to the right of <region1>?*
>
> **A:** In Figure 1 (response pdf), we show examples of SpatialRGPT handling multi-hop reasoning; we recommend zooming in for a clearer view. In the upper left sample, the model first identifies what's to the right of Region [0] (a single apple), finds the basket there, determines what's inside the basket, and then provides spatial details about the object inside. Even though our training data doesn't specifically include such multi-hop tasks, SpatialRGPT can still manage them effectively. This indicates that the model has developed a strong understanding of spatial relationships.

---

> > ### Comment · Reviewer_v4dm · 2024-08-12
> > **Thank you for your response**
> >
> > Thank you for your responses. They have adequately addressed my concerns regarding the data quality and its effects on the model itself. I hope the author can put the additional results in the revised version. Considering the value of this work, I will keep my original score.

---

### Official Review · Reviewer_hHf2 · 2024-07-12

**Soundness:** 3
**Presentation:** 3
**Contribution:** 3
**Rating:** 6
**Confidence:** 4

**Summary:**

This paper constructs region-aware spatial reasoning QA datasets from existing sources, resulting in the Open Spatial Dataset (OSD). Based on the OSD, they develop a model called SpatialRGPT, which integrates depth information to enable effective representation of regional information and acquisition of spatial knowledge. Experimental results demonstrate the superior spatial reasoning ability of the proposed model and its potential applications in robotics.

**Strengths:**

1. The proposed dataset OSD is well-crafted with open-vocabulary regions, which should benefit future research by enhancing models’ spatial reasoning abilities.
2. Integrating depth information into the VLM is a novel approach for achieving more accurate spatial reasoning.
3. The paper is well-written and easy to follow.

**Weaknesses:**

1. The SpatialRGPT-Bench is constructed through the proposed data generation pipeline, sharing the same answer formats as the OSD dataset on which the SpatialRGPT is trained. This may introduce bias when directly comparing it with other models not trained on the OSD dataset.
2. There is a lack of clarity regarding the size of SpatialRGPT used in the experiments. In Table 1, it is compared with a 34B model (LLaVA v1.6), while in Table 2, it is compared with 7B models.

**Questions:**

1. In Figure 4, the fifth QA example asks, "What kind of vehicles would not fit in?". Is this an example from the SpatialRGPT-Bench, and which category does it belong to? This question involves height, width, and depth of the garage area, which may not align well with the current categories and metrics used for evaluation.

**Limitations:**

See Weaknesses.

---

> ### Author Rebuttal · Authors · 2024-08-07
>
> **Q:** *SpatialRGPT-Bench is constructed through the proposed data generation pipeline, sharing the same answer formats as the OSD dataset on which the SpatialRGPT is trained. This may introduce bias when directly comparing it with other models not trained on the OSD dataset.*
>
> **A:** Please refer to the [General Response](https://openreview.net/forum?id=JKEIYQUSUc&noteId=04aWnh2vt2) for clarification. We explain the steps we have taken to ensure fair evaluation of SpatialRGPT-Bench in (A), conduct additional experiments on GPT-4 augmented SpatialRGPT-Bench in (B), and evaluate SpatialRGPT’s performance on a public benchmark, BLINK, in (C).
>
> ---
>
> **Q:** *There is a lack of clarity regarding the size of SpatialRGPT used in the experiments. In Table 1, it is compared with a 34B model (LLaVA v1.6), while in Table 2, it is compared with 7B models.*
>
> **A:** As mentioned in Section 3.3 (L209), we use LLaMA2-7B as our base LLM. We will include our model size in the tables for clarity. In Table 1 (submission), we compare our model to larger models (≥ 7B) in the spatial-related benchmark, while in Table 2  (submission), we compare it to models of the same size (7B) for general VLM benchmarks.
>
> ---
>
> **Q:** *In Figure 4, the fifth QA example asks, "What kind of vehicles would not fit in?". Is this an example from the SpatialRGPT-Bench, and which category does it belong to? This question involves height, width, and depth of the garage area, which may not align well with the current categories and metrics used for evaluation.*
>
> **A:** No, the questions in Figure 4 (submission) are not examples from the SRGPT-Bench. Examples from the SRGPT-Bench can be found in Figure 7 (submission), where each question contains only one category type (height, width, etc.).

---

> > ### Comment · Reviewer_hHf2 · 2024-08-10
> >
> > Thank you for your response, which has addressed my concern about the fairness in evaluation. I would like to raise the rating from 5 to 6.

---

### Official Review · Reviewer_ojLE · 2024-07-13

**Soundness:** 4
**Presentation:** 3
**Contribution:** 4
**Rating:** 7
**Confidence:** 4

**Summary:**

The paper introduces a novel approach for generating 3D, region-aware annotations from 2D images, transforming scene graphs into spatial QA training data for VLMs using a combination of template-based and LLM approaches. Key contributions include:
1. A novel pipeline for automatic generation of complex, metric spatial QA data.
2. A proposed depth adapter to include relative depth maps as input to the VLM.
3. Benchmarking the generated annotations against state-of-the-art methods, showing improvements in spatial reasoning tasks.

**Strengths:**

- The integration of scene graphs with template-based and LLM-based QA generation, along with the use of a depth map adapter, provides a novel approach to spatial QA VLM training.
- The experimental results are strong, demonstrating significant improvements over state-of-the-art models like GPT-4V+SoM and Llava-Next.
- The methodology is well-explained, with clear descriptions of the data collection process and the architecture of the proposed model.

**Weaknesses:**

- The paper lacks a detailed analysis of how closely the questions in the evaluation set match the templated questions from the data generation pipeline. It is unclear if the formatting provides an unfair advantage to the model, and whether altering the evaluation question formatting affects model performance.
- The paper does not include a discussion on whether the depth adapter alone, when trained on non-spatial data, improves performance. Additionally, it is unclear why SpatialVLM, a relevant baseline, was not included in the comparisons.

**Questions:**

1. How close are the questions in the evaluation set to the templated questions in the data generation pipeline? What steps are taken to ensure that the question and answer formatting do not give an evaluation advantage? If you change the evaluation question formatting, does it affect the model's performance?
2. Does the depth adapter addition alone, trained on non-spatial data, provide a performance boost even without the added spatial training data?
3. Why is SpatialVLM not included as a baseline, given its relevance?
3. What is the human performance on the evaluation set or a subset, given that humans may be worse than the model at metric spatial reasoning?

**Limitations:**

The authors discuss limitations in the appendix. Moving limitation discussion to the main paper and addressing the questions outlined above would strengthen the paper.

---

> ### Author Rebuttal · Authors · 2024-08-07
>
> **Q:** *How close are the questions in the evaluation set to the templated questions in the data generation pipeline?*
>
> **A:** The questions from both the evaluation set and data generation pipeline are randomly sampled from a set of templates.
>
> ---
>
> **Q:** *What steps are taken to ensure that the question and answer formatting do not give an evaluation advantage?*
>
> **A:** We have taken measures to avoid potential advantages from the QA format. Please see [General Response](https://openreview.net/forum?id=JKEIYQUSUc&noteId=04aWnh2vt2) (A) for clarification.
>
> ---
>
> **Q:** *If you change the evaluation question formatting, does it affect the model's performance?*
>
> **A:** In [General Response](https://openreview.net/forum?id=JKEIYQUSUc&noteId=04aWnh2vt2) (B), we conduct an experiment on a GPT-4 augmented SpatialRGPT-Bench. The results demonstrate that SpatialRGPT continues to outperform the baselines even when the questions and answers differ from the training data. Additionally, in [General Response](https://openreview.net/forum?id=JKEIYQUSUc&noteId=04aWnh2vt2)  (C), we show that SpatialRGPT is state-of-the-art on a public depth-related benchmark.
>
> ---
>
> **Q:** *Does the depth adapter addition alone, trained on non-spatial data, provide a performance boost even without the added spatial training data?*
>
> **A:** No. As mentioned in Line 223, the depth connector is a plugin module specifically trained on spatial-related QAs. Since it is not trained on non-spatial data and, thus, does not improve performance when spatial-related data is not available. For non-spatial data, depth inputs are not included to avoid redundant information from depth data for non-spatial tasks.
>
> ---
>
> **Q:** *Why is SpatialVLM not included as a baseline, given its relevance?*
>
> **A:** SpatialVLM is not open-sourced. In Table 4 (response pdf), we provide a comparison to SpaceLLaVA, a 3rd party community implementation mentioned on SpatialVLM’s website.
>
> ---
>
> **Q:** *What is the human performance on the evaluation set or a subset, given that humans may be worse than the model at metric spatial reasoning?*
>
> **A:** We show human performance on SpatialRGPT-Bench in Table 4 (response pdf). We observe that while qualitative QAs are easy for humans (97% average accuracy), quantitative QAs are extremely hard for humans (less than 50% accuracy). This supports the reviewer's suggestion that humans may be worse at metric-scale spatial reasoning.
>
> ---
>
> **Q:** *The authors discuss limitations in the appendix. Moving limitation discussion to the main paper and addressing the questions outlined above would strengthen the paper.*
>
> **A:** Thank you for the suggestion, we will revise the paper accordingly.

---

> > ### Comment · Reviewer_ojLE · 2024-08-10
> > **Thank you for your comments**
> >
> > Thank you to the authors for their comments and the additional experiments. My concerns have been addressed. The new experiments on rephrased questions and BLINK are valuable additions to the work. I have updated my score to a 7.

---

### Official Review · Reviewer_y3Gs · 2024-07-18

**Soundness:** 2
**Presentation:** 3
**Contribution:** 3
**Rating:** 6
**Confidence:** 5

**Summary:**

The paper introduces SpatialRGPT, a framework designed to enhance region-level spatial reasoning in Visual Language Models (VLMs) by incorporating 3D and region-aware visual encoder architecture. The authors present a scalable data pipeline to generate region-aware spatial reasoning questions and answers from existing datasets, resulting in the creation of the Open Spatial Dataset (OSD). To evaluate the model's performance, they introduce SpatialRGPT-Bench, a comprehensive benchmark with ground-truth 3D annotations. The paper demonstrates practical applications of SpatialRGPT, such as serving as a region-aware dense reward annotator for robotics and a stand-alone complex spatial reasoner.

**Strengths:**

- The paper addresses the important problem of enhancing the spatial perception capabilities of multimodal LLMs.

- It creates a large-scale training dataset with millions of examples.

- The paper is well-organized and easy to follow, clearly explaining the authors' motivations at each step.

- The effectiveness of the approach is demonstrated not only in vision-language tasks but also in embodied tasks.

**Weaknesses:**

- The biggest weakness, in my opinion, is the evaluation. The evaluation using SpatialRGPT-Bench shares the same data creation pipeline as the training data. This means the good performance on SpatialRGPT-Bench might just reflect the model learning the language style of the training data. Using GPT-4 for evaluation further biases the assessment towards responses that include numerical language, as seen in the teaser example: “The height of..1....is 204.54 feet. Assuming each floor is about 10 feet high, the total number of floors would be 20.454. Since you can’t have a fraction of a floor, the total number of floors would be approximately 20.” This type of response, while technically correct, doesn’t align with normal logical thinking. Therefore, a proper evaluation should be conducted on benchmarks like BLINK, especially those related to 3D tasks like depth.

- In Table 2, SpatialRGPT-Depth underperforms the original VILA in 6 out of 8 benchmarks.

- What if we use an off-the-shelf pretrained 3D detector on SpatialRGPT-Bench and then use an LLM to answer questions based on the cuboids from the 3D detector? On other benchmarks like BLINK, is SpatialRGPT better, or is the data curation method of first extracting 3D scene graphs and then using an LLM to summarize better?

- Does stage three of training (Visual Instruction-tuning) require updating all model parameters?

- Does the training data need to overlay all region proposals on the original images, like in Fig. 2, similar to Set-of-Marks?

- The model explanation is unclear, and Figure 3 is confusing. So, the input includes RGB, depth maps, and region proposals (masks or bounding boxes)? Then, a shared visual encoder extracts global features from the RGB image and depth map, and independent connectors project the global RGB/depth feature embeddings into the word embedding space? How many tokens represent RGB and depth, respectively? For region-level features, is each object represented by two tokens, one from the RGB feature and one from the depth feature? Does the Region Feature Extractor take features from the last layer of the visual encoder? Do the RGB and depth features share the same Region Feature Extractor? Figure 3 shows it as shared, but the appendix suggests they are independent.

- Why does the model need to extract region-level tokens separately? Couldn’t the region proposal information be included in the image-level token sequence through visual prompting, like in Set-of-Marks (SOM)? If LLAVA+SOM were trained on OSD, it might also work.

- The model size is not reported—Is it 7B?

- Typos: Line 327 should refer to Fig. 5.

**Questions:**

Please answer the questions in weakness section.

**Limitations:**

Please refer to the weakness section.

---

> ### Author Rebuttal · Authors · 2024-08-07
>
> **Q:** *SpatialRGPT-Bench uses same data pipeline as the training data, so its good performance might just reflect the model learning the training data's language style.*
>
> **A:** Please see General Response (B), we conduct additional experiments on a GPT-4 augmented SpatialRGPT-Bench. The results show that SpatialRGPT consistently outperforms the baseline models, even with different questions and answers from the training data.
>
> ---
>
> **Q:** *Using GPT-4 for evaluation biases the assessment towards responses that include numerical language, as seen in the teaser example: “...”...doesn’t align with normal logical thinking.*
>
> **A:** As mentioned in General Response (A), we employed in-context learning for baselines. With in-context learning, we found that GPT-4 is 100% and GPT-4V is 99% willing to provide answers consisting of numbers and units for all quantitative samples. Additionally, our quantitative benchmark only contains straightforward spatial questions, such as the width or height of an object. The teaser sample mentioned by the reviewer is not included.
>
> ---
>
> **Q:** *Evaluation should be conducted on benchmarks like BLINK, especially the depth task.*
>
> **A:** In General Response (C), we show SpatialRGPT’s results on BLINK’s Relative Depth Benchmark. **SpatialRGPT outperforms current SOTA with over 20% accuracy.**
>
> ---
>
> **Q:** *In Table 2, SpatialRGPT underperforms the original VILA.*
>
> **A:** In Table 2 of our submission, we demonstrate that SpatialRGPT maintains comparable performance on general VLM benchmarks, as adding new tasks often leads to a significant drop in the original model's performance. We further conduct a series of studies on SpatialRGPT with different model sizes from VILA-1.5 (3B and 8B), benchmarking it on general VLM, region understanding, and spatial benchmarks. The results in Table 2, Table 3, and Table 4 (response pdf) illustrate SpatialRGPT's ability to learn spatial capabilities without sacrificing performance on general and regional understanding benchmarks. **Notably, our 8B model shows consistent improvements compared to baseline, with more than a 2-point improvement on most benchmarks.**
>
> ---
>
> **Q:** *Can we use a 3D detector on SpatialRGPT-Bench and then use LLM to answer questions based on the cuboids from the 3D detector?*
>
> **A:** Following the reviewer's suggestion, we employ an Omni3D pretrained 3D detector and use GPT-4 to answer questions based on the detected cuboids. As shown in Table 4 of our response, LLMs struggle to effectively use coordinate information when presented in the text. Similar findings are reported in OpenEQA, where GPT-4 equipped with 3D bounding box information performs no better than without it.
>
> ---
>
> **Q:** *Compare SpatialRGPT vs LLM + data curation method on BLINK.*
>
> **A:** Our data curation method is object-centric, whereas BLINK requires point-level depth understanding. Therefore, our data pipeline cannot be directly applied to BLINK.
>
> ---
>
> **Q:** *Does stage three of training require updating all model param.?*
>
> **A:** No. We freeze the vision encoder and update the rest.
>
> ---
>
> **Q:** *Does the training data need to overlay all region proposals on the original images, like in Fig. 2, similar to SoM?*
>
> **A:** No. SpatialRGPT does not require overlayed region proposals on images. The overlay in Figure 2 (submission) is purely for visualization. Overlaying region proposals on images (as done in SoM) is straightforward but has significant drawbacks:
> * Ambiguity: The exact boundaries of the desired region are unclear.
> * Occlusion: Marks or lines can hide regions, hindering semantics, and small regions may be entirely obscured by annotations.
> * Sensitivity: The performance can be affected by the design of the marks, including shape, size, and color, as shown in prior studies.
>
> ---
>
> **Q:** *The model explanation and Fig. 3 is unclear. Does the input include RGB/D, and region proposals?*
>
> **A:** Yes.
>
> ---
>
> **Q:** *How many tokens represent RGB/D, respectively?*
>
> **A:** A shared visual encoder extracts 576 tokens each from the RGB image and depth map (24 * 24). Each RGB token is projected into the word embedding space through the RGB connector. Only the RGB tokens (576) are prepended before the text as a global context.
>
> ---
>
> **Q:** *For region-level features, is each object represented by two tokens, one from the RGB feature and one from the depth feature? Does the Region Feature Extractor take features from the last layer of the visual encoder?*
>
> **A:** Yes. The extractor takes 576 tokens from the last visual encoder layer. These are upsampled to 9216 tokens (96 * 96) in the Feature Refinement Layer. After the Mask-pooling Layer, each object has one RGB and one depth token, which are then projected into the word embedding space through separate connectors.
>
> ---
>
> **Q:** *Do the RGB and depth features share the same Region Feature Extractor? Figure 3 shows it as shared, but the appendix suggests they are independent.*
>
> **A:** The Region Feature Extractor consists of a Feature Refinement Layer (deconvolutions) and a Mask-pooling Layer (no parameters). RGB and depth features have separate Feature Refinement Layers. We will revise Figure 3 in the submission accordingly.
>
> ---
>
> **Q:** *Why does the model need to extract region-level tokens separately? Why not use visual prompting like in SoMs?*
>
> **A:** Using region extractors to obtain region-level tokens avoids SoM's drawbacks:
> * Ambiguity: Boxes or masks precisely identify regions.
> * Occlusion: No overlays mean regions aren’t obscured.
> * Sensitivity: No need for annotations, eliminating concerns about mark design.
>
> ---
>
> **Q:** *The model size is not reported—Is it 7B?*
>
> **A:** Yes. As mentioned in Section 3.3 (L209 submission) we use LLaMA2-7B as our base LLM. We will include this information in the main table.
>
> ---
>
> **Q:** *Typos: Line 327 should refer to Fig. 5.*
>
> **A:** Thank you for pointing out the typo. Line 327 should indeed refer to Figure 5 We will revise it accordingly.

---

> > ### Comment · Reviewer_y3Gs · 2024-08-11
> > **Thanks for the rebuttal**
> >
> > The additional evaluations provided by the author have addressed most of my concerns, and the author has also clarified the unclear parts of the paper. I hope the author can incorporate these details into the updated version to make the paper even better. I will raise my score to 6.

---

### Author Rebuttal · Authors · 2024-08-07

We thank the reviewers for recognizing the importance of our research problem (Reviewer `y3Gs`), and for acknowledging the novelty (Reviewer `ojLE`, `hHf2`), effectiveness (Reviewer `y3Gs`), and usefulness (Reviewer `v4dm`) of our approach. Below, we address the reviewers' common feedback, particularly regarding the evaluation of SpatialRGPT.

**(A) Ensuring Fair Evaluation on SpatialRGPT-Bench**

While Reviewer `y3GS`, `ojLE` and `hHf2` pointed out that the current benchmark may be biased to the proposed model, to ensure a fair evaluation on SpatialRGPT-Bench, we took the following measures to avoid potential advantages from the QA format:
* **Number-Aware Evaluations:** As mentioned in Line 738 of the submission, for quantitative questions, we used GPT-4 to extract numerical values and units from the answers. We calculated accuracy and error metrics only on these extracted values, ensuring that the evaluation did not favor our model's response due to textual formatting.
* **In-Context Learning for Baselines:** We provided baseline models with example QA samples, including both quantitative and qualitative questions, to enable in-context learning.

**(B) Augmented/Rephrased SpatialRGPT-Bench**

Following Reviewer `ojLE` and `hHf2`'s suggestions, we conduct additional experiments by augmenting and rephrasing both questions and answers in SpatialRGPT-Bench using GPT-4. The results are shown in Table 1 (response pdf). The results show that SpatialRGPT consistently outperforms the baseline models, even when the questions and answers are different from the training data.

**(C) Evaluation Results on BLINK**

Following Reviewer `y3Gs`'s suggestion, we evaluate SpatialRGPT on BLINK’s Relative Depth Benchmark. This benchmark is particularly challenging as it assesses point-level depths, while both the point-level region input and point-level questions were not specifically included in the training of SpatialRGPT. We use bounding boxes to mark the target points and evaluate the test set online with the EvalAI server. As shown in Table 6 (response pdf), SpatialRGPT significantly outperforms the state-of-the-art, **achieving over 20% accuracy gain compared to GPT-4V-Turbo**. Our model demonstrated strong performance, highlighting its ability to generalize to new tasks without explicit training.

---

### Decision · Program_Chairs · 2024-09-25

**Decision:**

Accept (poster)

**Comment:**

The reviewers unanimously vote for the acceptance of the paper. The rebuttal successfully addressed most of the concerns from the reviewers, including fairness in evaluation. The AC does not see any reason to turn down the reviewer opinions.